# Optimization of CRISPR/Cas9 Gene Editing System in Sheep (*Ovis aries*) Oocytes via Microinjection

**DOI:** 10.3390/ijms26031065

**Published:** 2025-01-26

**Authors:** Haitao Wang, Hengqian Yang, Tingting Li, Yan Chen, Jieran Chen, Xiaosheng Zhang, Jinlong Zhang, Yuting Zhang, Na Zhang, Runlin Ma, Xun Huang, Qiuyue Liu

**Affiliations:** 1State Key Laboratory of Molecular Developmental Biology, Institute of Genetics and Developmental Biology, Chinese Academy of Sciences, Beijing 100101, China; wanght@genetics.ac.cn (H.W.); yhq17806253397@163.com (H.Y.); tingli0704@163.com (T.L.); chenyan193@mails.ucas.ac.cn (Y.C.); 17852401863@163.com (J.C.); zhangyut09@163.com (Y.Z.); zhangna4520@163.com (N.Z.); rlma@genetics.ac.cn (R.M.); xhuang@genetics.ac.cn (X.H.); 2College of Animal Science and Technology, Qingdao Agricultural University, Qingdao 266109, China; 3Institute of Animal Sciences and Veterinary, Tianjin Academy of Agriculture Sciences, Tianjin 300381, China; zhangxs0221@126.com (X.Z.); jlzhang1010@163.com (J.Z.); 4Tianjin Key Laboratory of Animal Molecular Breeding and Biotechnology, Tianjin 300381, China

**Keywords:** CRISPR/Cas9, microinjection, gene editing, sheep

## Abstract

The CRISPR/Cas9 system has become a powerful tool for molecular design breeding in livestock such as sheep. However, the efficiency of the Cas9 system combined with zygote microinjection remains suboptimal. In this study, mature sheep oocytes were used for microinjection to assess the impact of various factors on Cas9 editing efficiency. We found that the in vitro maturation efficiency of oocytes is related to environmental factors such as air temperature, pressure, and humidity. Our results indicate that high-efficiency gene editing can be achieved when targeting the *SOCS2*, *DYA*, and *TBXT*, using a microinjection mixture with a concentration of 10 ng/μL Cas9 and sgRNA. By optimizing the injection capillary, we significantly reduced the oocyte invalidation rate post-microinjection to 3.1–5.3%. Furthermore, we observed that using either Cas9 protein or mRNA in the microinjection process resulted in different genotypes in the edited oocytes. Importantly, parthenogenetic activation did not appear to affect the editing efficiency. Using this high-efficiency system, we successfully generated *SOCS2* or *DYA* gene-edited sheep, with all lambs confirmed to be genetically modified. This study presents a highly efficient method for producing gene-edited sheep, potentially enabling more precise and effective strategies for livestock breeding.

## 1. Introduction

CRISPR/Cas9 (Clustered regularly interspaced short palindromic repeats/CRISPR-associated 9) is the most widely used gene editing tool. The CRISPR/Cas system can defend against viral invasion in archaea by editing invading foreign genes [1]. Harnessed as a gene editing tool, the Cas9 protein binds with sgRNA (small-guide RNA) sequence, which guides the Cas9/sgRNA to the target site of the genome [2]. The HNH and RuvC endonuclease domain can conduct DNA cleavage, including the complementary and non-complementary strands of the DNA helix [3]. Deletion and insertion of bases at the target site were produced through DNA repair, thereby achieving the editing of the target genes. The CRISPR/Cas9 system has been used in genome design breeding across various livestock species, including sheep (*Ovis aries*) [4]. Molecular breeding techniques have been increasingly applied in sheep farming by using target genes associated with economic traits. The Cas9 can perform precise editing in animals by microinjection of zygotes, followed by embryo transfer to produce genetically edited sheep [5]. Targeted major gene is usually related to economic performance, such as growth rate and disease resistance [6]. Achieving high-efficiency editing in zygotes is crucial for increasing the success rate of generating edited sheep and reducing costs. However, various conditions and factors can influence the efficiency of Cas9 editing in sheep via microinjection, and it is necessary to optimize these conditions.

Previous studies have attempted to generate gene-edited sheep through microinjection, but the editing efficiency has not been satisfactory. Elevating the efficiency of microinjection in sheep is rarely reported. *SOCS2* (suppressor of cytokine signaling 2) is associated with animals’ growth performance by regulating the GH/IGF1 (growth hormone/insulin-like growth factor 1) axis [7,8]. *DYA* (MHC class II antigen DY alpha) contribute to the adaptive immunity of sheep and may be related to resistance of nematodosis [9,10]. *TBXT* (T-box transcription factor T) is associated with tail length among sheep populations and vertebrae variation in mammals [11,12]. These genes have been previously utilized as target genes for generating gene-edited sheep models in our group. In this study, we optimized the efficiency of zygote editing via microinjection, specifically targeting the *SOCS2*, *DYA*, and *TBXT* genes.

To identify optimal conditions for high editing efficiency, we performed in vitro maturation of sheep oocytes and used these mature oocytes as a model for zygote editing via microinjections. We developed and implemented an optimized microinjection system, which demonstrated significantly enhanced efficiency in sheep gene editing. This system provides a promising approach for generating gene-edited sheep with greatly improved editing success.

## 2. Results

### 2.1. In Vitro Maturation Rate of Sheep Oocytes Influenced by Seasonal Variations

In this study, sheep oocytes matured in vitro were collected for microinjection. In order to increase the number of oocytes available for microinjection, we raised the concentration of luteinizing hormone (LH) and follicle-stimulating hormone (FSH) in the in vitro maturation (IVM) medium from 0.05 IU to 0.1 IU. Additionally, fetal bovine serum (FBS) was replaced with estrus ovine serum (EOS) in the IVM process. As a result, the oocyte maturation rate increased from 73.19% ± 10.03% to 79.10% ± 8.45% (Figure 1a), and the number of oocytes suitable for microinjection significantly rose from 42.44% ± 7.93% to 70.14% ± 8.31% (Figure 1b). Further analysis revealed that the oocyte maturation rate varied across different months and seasons, with higher rates observed in spring and autumn, and lower rates in summer (Figure 1c,d). Our experiments were conducted in Beijing, located at a latitude of 41° N in northern China, a region characterized by four distinct seasons. We reviewed the climate data for Beijing, which included monthly averages for sunlight duration, air temperature, relative humidity, and atmospheric pressure. These climate variables also exhibited seasonal fluctuations (Appendix A). A Spearman correlation analysis between oocyte maturation rate and these climate parameters revealed a negative association between maturation rate and both sunlight duration and air temperature, while relative humidity showed a positive correlation. No significant correlation was found between oocyte maturation rate and atmospheric pressure (Figure 1e–h, Appendix A). The climate conditions in summer were identified as the primary factors influencing maturation.

### 2.2. Optimization of CRISPR/Cas9 Editing Microinjection System in Oocytes

In this study, commercial microinjection capillaries were used for microinjection (Figure 2a) with a diameter of 0.5 μm capillary and a sloping injection during microinjection. The mortality rate of oocytes in 24 h after injection was as high as 22.53% ± 6.38%. To lower the mortality rate, new microinjection capillaries with a 0.25 μm diameter and a 20–30° bend were developed (Figure 2b). The capillaries could be horizontally stabbed into the oocyte during injection. After using the new capillaries, the mortality rate decreased to 5.28% ± 3.92% (Figure 2c).

The concentration of sgRNA and Cas9 protein in the injection mixture may affect the efficiency of gene editing. In order to obtain the optimal concentration of sgRNA or Cas9 for gene editing efficiency, a couple of sgRNA for SOCS 2 and DYA were used for microinjection, and a concentration gradient was designed for gene editing. In addition, GFP mRNA was used as an indicator for co-injection. After injection, oocytes without GFP fluorescence were removed, and the gene editing efficiency targeted to *SOCS2* was detected after the embryos developed to the morula stage. The results showed that *SOCS2* was edited by 1 ng/μL sgRNA and Cas9 protein successfully, but the gene editing efficiency was low (42.41% ± 27.28%). When the concentrations of sgRNA and Cas9 protein were 10 ng/μL or higher, the *SOCS2* gene editing efficiency could reach nearly 100% (Figure 3a). At the same time, there was no significant difference in the mortality rate and cleavage rate between the injected and non-injected oocytes (Figure 3b,c). Furthermore, we found that after the injection of GFP mRNA, the oocytes displayed different fluorescence intensities (Appendix A). By detecting the gene editing efficiency (targeted to *TBXT*) of oocytes with significantly different fluorescence intensities, there was no difference in editing efficiency between oocytes with higher and lower fluorescence intensity (Figure 3d), while the cleavage rate also significantly increased (Figure 3e).

Cas9 mRNA, combined with sgRNA, can also be used for gene editing. This study compared the gene editing efficiency of commercial Cas9 protein and mRNA. When targeting *SOCS2*, the average editing efficiency was 74.28% ± 11.16% with mRNA, and the homozygous editing efficiency was 48.89% ± 1.92%. In contrast, the use of Cas9 protein resulted in an average editing efficiency of 90.63% ± 10.97% and a homozygous efficiency of 79.41% ± 13.75% (Figure 4a,b). For the *TBXT*, the average editing efficiency with mRNA was 62% ± 20.26%, with a homozygous efficiency of 3.33% ± 5.77%. When using Cas9 protein, the average editing efficiency was 76% ± 16.27%, and the homozygous efficiency was 43.41% ± 30.46% (Figure 4c,d) (Appendix A). While the overall editing efficiency did not show significant differences between the two methods, there was a notable difference in homozygous editing rates for *SOCS2* (Appendix A).

To investigate the influence of parthenogenetic activation on the gene editing efficiency of oocytes, microinjection before and after parthenogenetic activation was performed, and editing efficiency was calculated. The results showed that the overall efficiency targeted to *SOCS2* was 94.74% ± 4.71% with microinjection before parthenogenetic activation, and the gene editing efficiency was 95.24% ± 8.25% with microinjection after parthenogenesis activation. There was no significant difference between editing efficiencies of microinjection before or after parthenogenetic activation (Figure 5a), indicating that gene editing efficiency is not affected by parthenogenetic activation. In addition, the cleavage rate and blastocyst rate after parthenogenetic activation were not significantly different from those before activation (Figure 5b).

### 2.3. The Development of Early Embryos Was Not Influenced by Microinjection

To detect the effect of microinjection on embryo development in this experiment, the development rate and blastocyst rate after the injection were analyzed. The results showed that the 48 h cleavage rate of the injected embryos was 74.87% ± 1.31%, while the 48 h cleavage rate of the non-injected embryos was 70.73% ± 3.02% (Figure 6a,b). The marker genes *OCT4*, *SOX2*, and *CDX2* in the blastocyst stage were detected by immunofluorescence, and the expression of the marker genes was not influenced by the injection (Figure 6c). In addition, Cas9 proteins were also detected in early embryos after injection by immunofluorescence; Cas9 can keep activity to the morula stage of the embryo, but not in the blastocyst stage (Figure 6d). The result indicated that the mRNA and the Cas9 protein are probably degraded over time.

### 2.4. High-Efficiency of CRISPR/Cas9 Editing in Sheep by Microinjection

To access the efficiency of this editing system developed in this study in sheep, *SOCS2* and *DYA* were targeted using co-injected sgRNA and Cas9 protein in zygotes obtained from ewes by superovulation. Gene-edited sheep were generated by embryo transfer. A total of 193 sheep zygotes were obtained from the donors. Forty-seven zygotes injected with *SOCS2* sgRNA were transferred to 7 recipient ewes, resulting in the natural delivery of 6 live lambs after 150 days of pregnancy. Editing of *SOCS2* was confirmed in all 6 lambs by TA cloning and Sanger sequencing using designed primers (Figure 7a,c) (Table 1) (Appendix A). The amino acid sequences of proteins were also predicted, and the edited SOCS2 showed truncated proteins (Table 2). Additionally, 90 zygotes injected with *DYA* sgRNA were transplanted to 15 recipients, leading to the natural delivery of 5 live lambs, with editing of *DYA* detected in all 5 lambs. (Figure 7b,d) (Table 1). Edited DYA proteins were also truncated (Table 2). Next generation sequencing (NGS) was conducted using ear skin DNA samples. All analyzed results of genotypes and ratios in each sample in target regions have been shown in Appendix A and Figure 8c,d. These results showed that all born lambs were gene-modified.

## 3. Discussion

Zygote microinjection involves injecting nucleic acids and Cas9 protein or mRNA into zygotes by the transplantation of the edited zygotes into surrogates to generate offspring. Compared to somatic cell nuclear transfer (SCNT), microinjection is faster, more cost-effective, and simpler in execution [13]. It also results in higher surrogate pregnancy rates and higher offspring survival rates. However, since gene editing occurs directly in the zygotes, it is necessary to identify the offspring’s genotype after birth. The success of generating edited animals is highly dependent on the efficiency of editing in zygotes, meaning that improving the gene editing process in zygotes will result in more gene-edited offspring. In this study, through the exploration of the component concentration in the injection mixture, the optimization of injection capillary, the comparison of injection before and after parthenogenetic, and the comparison of Cas9 protein or mRNA, an optimized system of CRISPR/Cas9 editing in sheep oocytes via microinjection was constructed. The editing efficiency targeted to *SOCS2* and *DYA* in sheep by microinjection of zygotes is 100%, which is higher compared to previous studies (5.7–97%) (Table 3) [14,15,16,17,18,19,20,21,22,23]. Our study provides a method for improving gene editing efficiency when generating gene-edited sheep by microinjection.

This study found that using at least 10 ng/μL Cas9 protein and sgRNA during microinjection can lead to high-efficiency editing in oocytes. For zygote microinjection, the concentration of Cas9 protein and sgRNA can be adjusted to exceed 10 ng/μL in the injection mixture. In gene editing of somatic cells or oocytes, vectors expressing both Cas9 protein and sgRNA can be used for liposome transfection or electroporation. Then, the Cas9 protein and sgRNA expressed to edite the target gene. However, compared to plasmid transfection, transfection or injection of Cas9 protein and sgRNA can directly target the gene of interest without the possibility of integrating into the genome, making it faster and safer. The original Cas9 system relied on Cas9 protein, CRISPR-derived RNA (crRNA), and trans-activating RNA (tracrRNA). Jinek et al. simplified the system by linking crRNA and tracrRNA [2]. Until now, the injection mixture only comprised Cas9 protein and sgRNA without any other components. Therefore, the concentration of the injection components is one of the critical factors that can affect gene editing efficiency. Currently, many commercialized Cas9 proteins can be used for microinjection, including Cas9 and cpf1 with different elements such as nuclear localization signals, tag proteins, and modifications to enhance its stability and safety (Appendix A) [24]. Those companies that developed these Cas proteins declared that all products have been laboratory-verified with high editing efficiency and stability. However, when used in different species or target genes, the editing efficiency of these proteins still needs to be detected, and it is necessary to select appropriate commercial proteins according to different species and genes.

We optimized gene-editing efficiency using in vitro matured embryos. Through parthenogenetic activation, in vitro matured oocytes can develop to the blastocyst stage without fertilization [25]. Therefore, the editing efficiency of mature oocytes can reflect that of early embryos. For target genes, multiple sgRNAs were designed, and high-efficiency sgRNAs were selected to generate edited animals. Assessing gene-editing efficiency in oocytes serves as a reference for selecting optimal sgRNAs for producing gene-edited animals. Our study found that oocyte maturation is influenced by seasonal climate conditions. Most sheep breeds exhibit seasonal estrus, largely due to variations in daylight duration. The climate at our experimental location (about 40° N, 116° E) is highly seasonal, and the oocyte maturation rate varied accordingly. This suggests that in vitro maturation may be affected by factors like daylight duration, temperature, and humidity, which align with reports in other species [26]. This result demonstrated that shorter sunlight duration, lower air temperature, and higher relative humidity of environment are beneficial for increasing oocyte maturation rate, especially for the relative humidity.

We also observed that using Cas9 protein or mRNA led to different gene-editing outcomes for the *TBXT* but not for *SOCS2*. This discrepancy might be due to differences in mRNA translation after injection, as well as the unique characteristics of each target gene and its sgRNAs. Commercialized Cas9 proteins were more stable compared to mRNA, which requires translation after injection.

Additionally, we investigated the influence of parthenogenetic activation on gene editing efficiency. Only unfertilized mature oocytes require parthenogenetic activation for development. This process involves elevating the Ca^2+^ levels in MII oocytes through physical or chemical stimulation: In this study, ionomycin was used for chemical activation, which leads to the decline or inactivation of factors such as metaphase-promoting factor (MPF) and cytostatic factor (CSF), enabling the oocytes to enter the meiosis II period and further cleavage [27]. The microinjection injection mixture consisted of purified Cas9 protein, sgRNA, and nuclease-free water. In our study, the results also showed that injections performed before or after activation do not impact development rates or editing efficiency. The results implied that the injection mixture did not affect the Ca^2+^ level of oocytes and parthenogenetic activation, and activation does not affect the editing process. However, in zygote microinjection, in vivo or in vitro fertilized oocytes were used, parthenogenetic activation is not a concern. Nevertheless, injections should be performed as soon as possible after fertilization to avoid editing after cleavage, which could result in genetic chimerism.

The results of TA cloning and NGS sequencing both indicated that all live animals underwent gene editing, and gene fragment disruptions were clearly identified. However, there are significant discrepancies between the results of TA cloning and NGS sequencing. The possible reasons are as follows: for TA cloning, even colonies were picked randomly as much as we could for sequencing. Some variants with lower frequencies may not be detected. In the meantime, the NGS sequencing has a higher throughput and depth. More variants present in the sample could be identified by NGS sequencing. Moreover, NGS technologies are limited by their short length of reads, usually paired ends of 150 bp, making it hard to accurately detect large deletions after DSBs [28]. Compared to somatic cell nuclear transfer, zygote microinjection combined with the CRISPR/Cas9 system is of high efficiency that allows direct embryo editing by a one-step embryo manipulation [29]. However, during the microinjection process, it is crucial to avoid injecting sgRNA and Cas9 after cleavage to minimize the risk of producing genetic mosaic individuals. Despite these precautions, the resulting lambs still require high-depth sequencing to assess whether they are genetically mosaic. Therefore, for molecular breeding based on gene modification, it is essential to select the F1 generation of gene-edited individuals, derived from the germline-edited F0, for further breeding. The generation of edited animals is influenced by multiple factors. Such as the design of sgRNAs or Cas9 editing efficiency, improving the quality of donor embryos and optimizing the estrous cycle of recipient ewes. Reducing mortality post-microinjection improves zygote utilization rates which can reduce the number of donors needed for recipients. While our high-efficiency editing system significantly advances the generation of gene-edited sheep, it does not guarantee 100% editing efficiency across all genes. Nevertheless, it provides a valuable framework for optimizing gene-edited animal production via zygote microinjection.

## 4. Materials and Methods

### 4.1. Animals and Ethical Statement

All sheep used for embryo transfer were Hu sheep and were bred by the Institute of the Animal Sciences and Veterinary, Tianjin Academy of Agriculture Sciences, Tianjin, China.

### 4.2. sgRNA and Cas9 mRNA or Protein

An sgRNA template sequence was designed according to the conference sequence of *SOCS2* (Oar v4.0, NC019460.2), *DYA* (Oar v4.0, NC019477.2), and *TBXT* (Oar v4.0, NC 019465.2). SgRNAs were listed in Appendix A and synthesized by Genescript (Nanjing, China), and sgRNAs were diluted with RNase-free water to 1 μg/μL. Cas9 mRNA (eSpCas9 mRNA, MA13810). GFP mRNA (eGFP mRNA, SC2325) was purchased from Genscript (Nanjing, China) and Cas9 protein was purchased from Invitrogen (Shanghai, China) (TrueCut Cas9 v2 A36497).

### 4.3. Manufacture of Capillaries for Microinjection

Capillary glass (Sutter Instrument Company, Wuhan, China, BF-100-78-10) is pulled by a micropipette puller P-2000 under the conditions of ramp 469, heat 465, pull 130, voltage 25, time 140, and a pressure of 500 to produce a micropipette. The micropipette capillary was bended to 20–30° by microforge MF-2 under a procedure of 65 °C and 2 s duration.

### 4.4. In Vitro Maturation of Sheep Oocytes

Cumulus-oocyte complexes (COCs) were separated from purchased Hu sheep or Small-tail Han sheep ovaries and transferred into in vitro mature medium including 0.2 mM Na-Pyvate (Sigma, St. Louis, MO, USA, P3662), 1 mM L-Glutamine (Sigma G3126), 0.1 IU FSH (NSHF), 0.1 IU LH (NSHF), 20 ng/mL EGF (Sigma E9644), 100 μM Cys (Sigma, C7352), 1 μg/mL β-E2 (Sigma, E8875), 10% FBS or estrus ovine serum, and M199 (Gibco, Grand Island, NY, USA, 11150-059). In brief, Syringes with 0.8 mm diameter needles were used for extraction of COCs from 2–4 mm follicles. The extracted COCs were collected under stereo microscope and washed by IVMs 3 times. Living COCs were picked and cultured in IVM under 38.5 °C, 5% CO_2_ for 20 h, 75–100 oocytes for every culture droplet. More than 100 oocytes were performed the IVM every time. The matured oocytes were incubated in 1% hyaluronidase for 5 min to remove cumulus cells. Oocytes with the first polar body were selected for microinjection.

### 4.5. Microinjection

The injection mixture consisted of sgRNA (1–100 ng/μL), Cas9 protein or mRNA (1–100 ng/μL), GFP mRNA (100 ng/L), and RNAse-free water, 10 μL in total. For GFP fluorescent intensity quantity, ImageJ was used for density analysis of every oocyte. Then, the average intensity was calculated. The oocytes with intensity higher than average were classified into a higher group, while the oocytes with intensity lower than average were classified into a lower group. Oocytes with higher and lower GFP densities were classified into groups to calculate the editing efficiency and cleavage rate of each group. GFP mRNA was removed from the mixtures for zygote microinjection. The mixture was put into the front end of the capillary (Eppendorf, Hamburg, Germany, 22290012 or developed artificially) and injected into the cytoplasm of mature oocytes or zygotes under pressure of 20–50 hPa and for a 1 s duration controlled by eppendorf FemtoJet 4i. Injected oocytes or zygotes were recovered in IVM for 30 min. More than 100 oocytes were performed to microinjection in every experiment.

### 4.6. Parthenogenetic Activation and In Vitro Development

After injection, oocytes were incubated in 5 μM ionomycin for 5 min (Merck, Darmstadt, Germany). Then, the oocytes were transferred into 2 mM 6-DAMP (Sigma-D2629) for 2.5–4 h. Activated oocytes were transferred into G1 (Vitrolife, Gothenburg, Sweden) for 48 h and into G2 (Vitrolife, Gothenburg, Sweden) for 3–5 days under 38.5 °C, 5% CO_2_, 5% O_2_, and 90% N_2_.

### 4.7. Immunofluorescence

Early embryos were incubated in acid Tyrode’ solution (LEAGENE, Beijing, China) for 3 min and fixed for 15 min in 4% paraformaldehyde. After permeabilizing samples, the embryos were treated for 10 min with 0.1% Triton X-100 in PBS and blocked with 1% BSA in PBS. Samples were incubated with primary antibodies overnight at 4 °C and secondary antibodies for 1 h at 37 °C. The anti-OCT4 antibody (11263) was purchased from Proteintech (Wuhan, China); the anti-SOX2 antibody (365823) was purchased from Santa Cruz (Shanghai, China); the anti-CDX2 antibody (MU392A-5UC) was purchased from BioGenex (Fremont, CA, USA). The anti-Cas9 antibody (14697S) was purchased from Cell Signaling Technology (Danvers, MA, USA). A secondary antibody marked with CY3 (GB21301) was purchased from Servicebio (Wuhan, China). A secondary antibody marked with Alexa Fluor 488 (A11008) was purchased from Invitrogen. Negative immunofluorescence controls were performed by using the same method without a specific antibody. Finally, samples were stained with DAPI in fluoromount-G (Beyotime, Shanghai, China, P0131) for 10 min and covered with mineral oil. Antibody binding was viewed with a laser-scanned Leica SP8 confocal microscope.

### 4.8. Statistical Analysis

The meteorological data used in this study were downloaded from the National Meteorological Science Data Center, China Meteorological Administration (https://data.cma.cn/ (accessed on 24 June 2024)). For quantitative data, if the distribution of different groups meets the normal distribution, a one-factor analysis of variance (ANOVA) is used. The LSD (Least Significant Difference) LSD test is used when the variance is homogeneous, and the Games–Howell test is used when the variance is not homogeneous. If the distribution does not meet the normal distribution, the Kruskal–Wallis H test is used for multiple comparisons. For the correlation analysis between two groups of quantitative data, if both groups meet the normal distribution, the Pearson product-moment correlation coefficient is used; otherwise, the Spearman rank correlation coefficient is used.

### 4.9. Generation of Gene-Edited Sheep

Sheep zygotes were microinjected as follows: The injection mixture consisted of sgRNA (1–100 ng/μL), Cas9 protein, and RNAse-free water, 10 μL in total. The mixture was injected into the cytoplasm of zygotes under pressure of 20–50 hPa and for a 1 s duration controlled by eppendorf FemtoJet 4i. After injection, zygotes were transferred into the oviduct of synchronized estrous recipient sheep. We monitored the pregnancy status of the recipient for 142–155 days until delivery. All lambs were delivered naturally.

### 4.10. Genotyping of Edited Sheep

For early embryos, morula and blastocysts were selected and lysed according to the protocol of EZ-editor Lysis Buffer (Ubigene Biosciences, Guangzhou, China) to obtain genomic DNA (gDNA). For lambs, venous blood or ear tissues of the newborn lambs was collected, and gDNA was isolated according to the protocols of the TIANGEN blood/cells/tissue DNA extraction kit (TIANGEN, Beijing, China, DP304-03) for genotyping: gDNA of embryos or lambs was used as a template for amplification of target genes. Primers of the target gene are shown in the Appendix A. The PCR procedure is as follows: 95 °C for 1 min, followed by 35 cycles of 95 °C for 10 s, 59 °C for 15 s, 72 °C for 15 s, and 72 °C for 5 min. PCR products underwent TA cloning by using the Hieff Clone Zero TOPO-TA Cloning Kit (YEASEN, Shanghai, China, 10907ES), and genotypes of colonies were sequenced by Sango Biotech (Shanghai, China). Genome DNA of blood were prepared for targeted amplicon sequencing. In brief, 100 ng of genomic DNA was added in a 25 μL PCR reaction with 0.2 μM primers and amplified for 35 cycles. The PCR products were subjected to paired-end sequencing on the Hi-TOM platform of XI’AN CyanSnow Gene Company (Xi’an, China), generating 150 base pair (bp) reads for each end of the DNA fragments, and they were assembled with the reference sequence. All analyzed results of genotypes and ratios in each sample on target regions were shown in stacked bar charts. The raw sequencing reads were aligned to the reference genome using the BWA-MEM algorithm. Then, the alignment files in the BAM format were produced, and these contain the mapped reads and their corresponding genomic positions. The generated BAM files were further processed to improve data quality. First, the aligned reads were sorted by genomic coordinates using SAMtools (V1.16.1). Subsequently, PCR duplicates were marked using Picard Tools (V3.0) to identify potential biases introduced during PCR amplification. The Integrative Genomics Viewer (IGV) was employed to inspect the mapped reads.

## 5. Conclusions

In this study, in vitro maturation sheep oocytes were used for zygote editing via microinjection, and the in vitro maturation efficiency of oocytes is influenced by environmental factors. An optimized microinjection system was developed, and high editing efficiency of target genes was detected in oocytes, and all generated lambs were detected to be genome-modified.

## Figures and Tables

**Figure 1 ijms-26-01065-f001:**
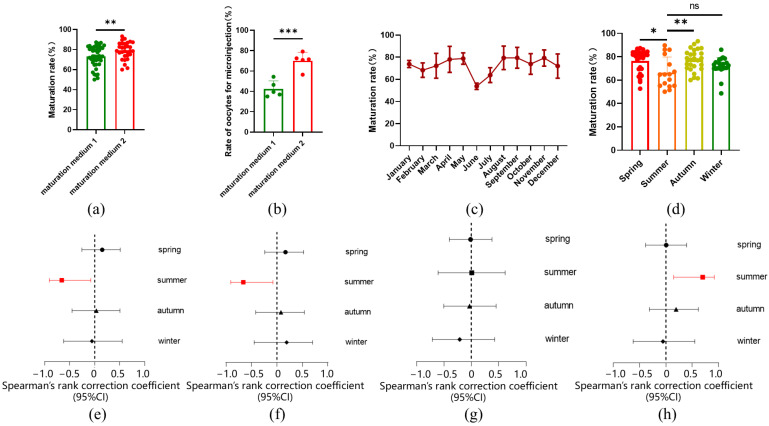
IVM rates of sheep oocytes. (**a**) IVM rates of oocytes with increased hormone concentration and serum EOS (maturation medium 1: *n* = 46; maturation medium 2: *n* = 30). The rate of IVM is ratio of living oocyte with first polar body numbers to numbers of all living oocytes; (**b**) Proportion of available oocytes with increased hormone concentration and serum EOS (maturation medium 1: *n* = 5; maturation medium 2: *n* = 5). The rate of oocytes for microinjection is the ratio of oocytes numbers for microinjection to numbers of all mature oocytes; (**c**) IVM rate of oocytes varied over different months (January: *n* = 9; February: *n* = 5; March: *n* = 9; April: *n* = 9; May: *n* = 9; June: *n* = 5; July: *n* = 5; August: *n* = 6; September: *n* = 11; October: *n* = 11; November: *n* = 4; December: *n* = 8); (**d**) IVM rate of oocytes varied with different seasons (Spring: *n* = 27; Summer: *n* = 16; Autumn: *n* = 26; Winter: *n* = 22), the ns means not significant; (**e**) Forest plot of air temperature on the maturation rate of oocytes in different seasons; (**f**) Forest plot of sunlight duration on the maturation rate of oocytes in different seasons; (**g**) Forest plot of atmospheric pressure on the maturation rate of oocytes in different seasons; (**h**) Forest plot of relative humidity on the maturation rate of oocytes in different seasons. All data are presented as mean ± standard deviation (SD). * *p* < 0.05, ** *p* < 0.01, *** *p* < 0.001.

**Figure 2 ijms-26-01065-f002:**
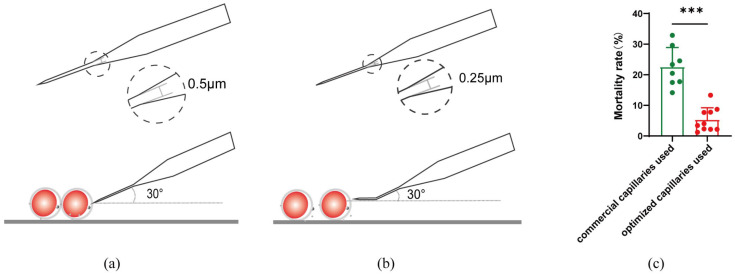
Reduced mortality with horizontal injection by optimized capillary. (**a**) A commercial microinjection capillary with a diameter of 0.5 μm, stabbing into the oocyte slantly; (**b**) Optimized microinjection capillary with a diameter of 0.25 μm, stabbing into the oocyte horizontally; (**c**) Reduced mortality by using different injection capillaries. All data are presented as the mean ± SD. Commercial capillaries used: *n* = 8, Optimized capillaries used: *n* = 10. *** *p* < 0.001.

**Figure 3 ijms-26-01065-f003:**
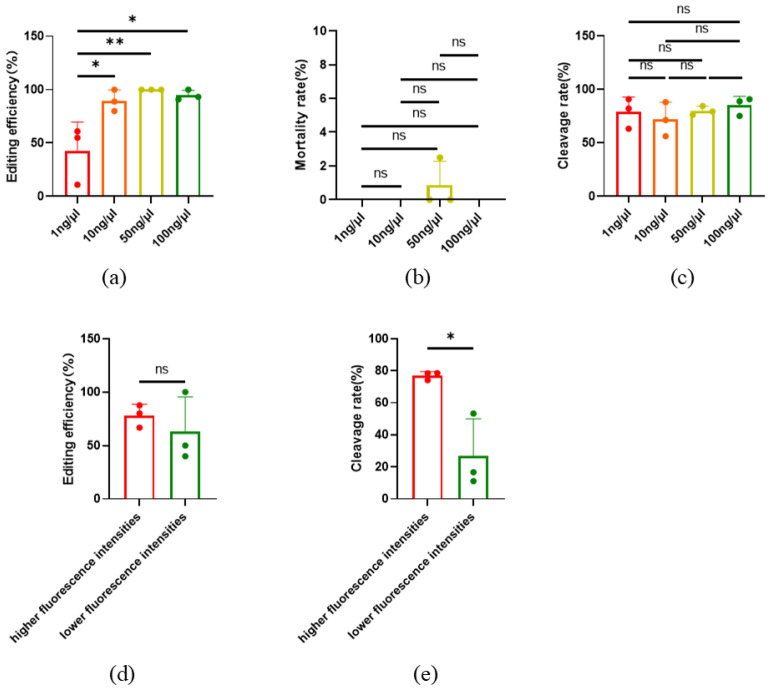
Gene editing efficiency under different concentrations of injection mixture. (**a**) Editing efficiency at different injection concentrations, *n* = 3; (**b**) Death rate of oocytes at different injection concentrations. *n* = 3. The ns means not significant; (**c**) Cleavage rate at different injection concentrations, *n* = 3. The ns means not significant. (**d**) Editing efficiency at different fluorescence intensities, Higher: *n* = 3, lower, *n* = 3. All oocytes here expressed GFP. The ns means not significant; (**e**) Cleavage rate at different fluorescence intensities. Higher: *n* = 3, lower: *n* = 3. All oocytes here expressed GFP. All data are presented as the mean ± SD. * *p* < 0.05, ** *p* < 0.01.

**Figure 4 ijms-26-01065-f004:**
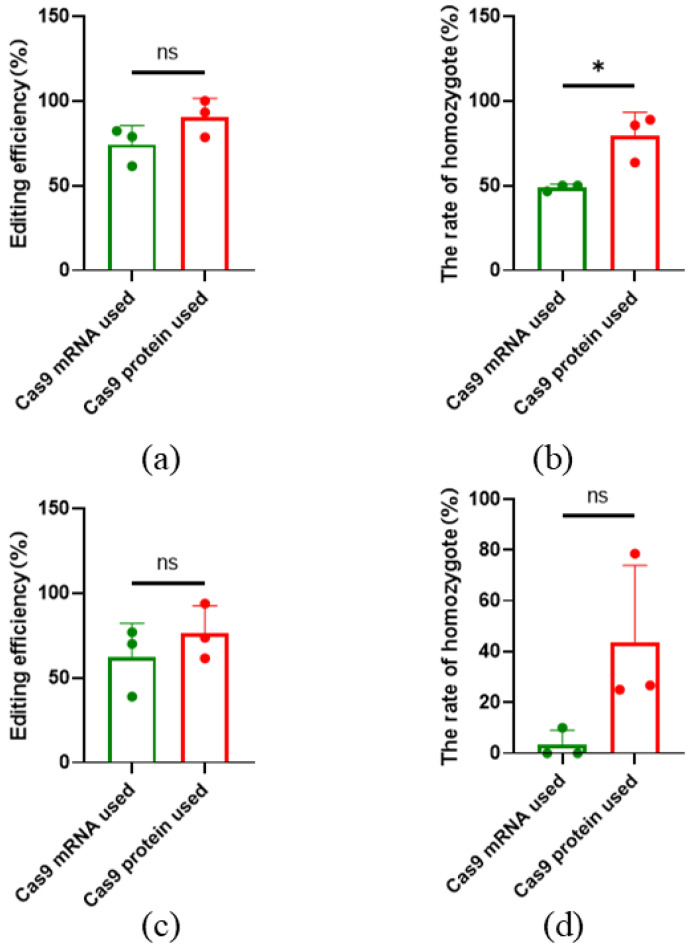
Effect of using Cas9 mRNA and protein on gene editing results. (**a**) Gene editing efficiency for *SOCS2*, the ns means not significant; (**b**) Homozygous editing efficiency for *SOCS2*; (**c**) Gene editing efficiency for *TBXT*, the ns means not significant; (**d**) Homozygous editing efficiency for *TBXT*. All data are presented as the mean ± SD. *n* = 3. * *p* < 0.05, the ns means not significant.

**Figure 5 ijms-26-01065-f005:**
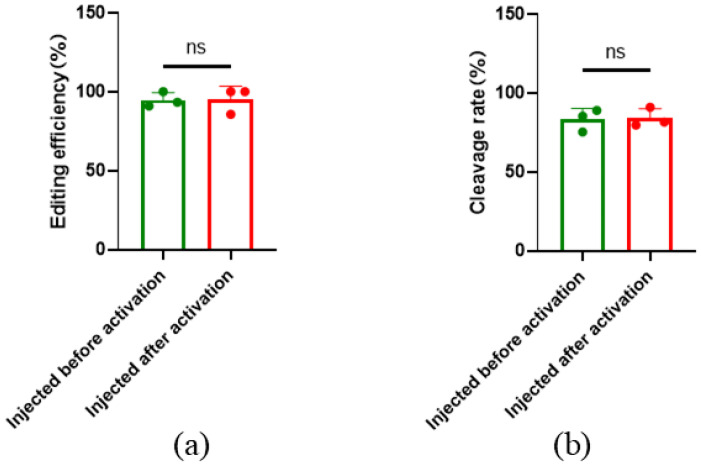
*SOCS2* editing efficiency before and after parthenogenetic activation. (**a**) Gene editing efficiency before and after parthenogenesis activation, the ns means not significant; (**b**) Cleavage rate before and after parthenogenesis activation. Before: Injected before parthenogenetic activation; after: Injected after parthenogenetic activation. All data are presented as the mean ± SD, *n* = 3, the ns means not significant.

**Figure 6 ijms-26-01065-f006:**
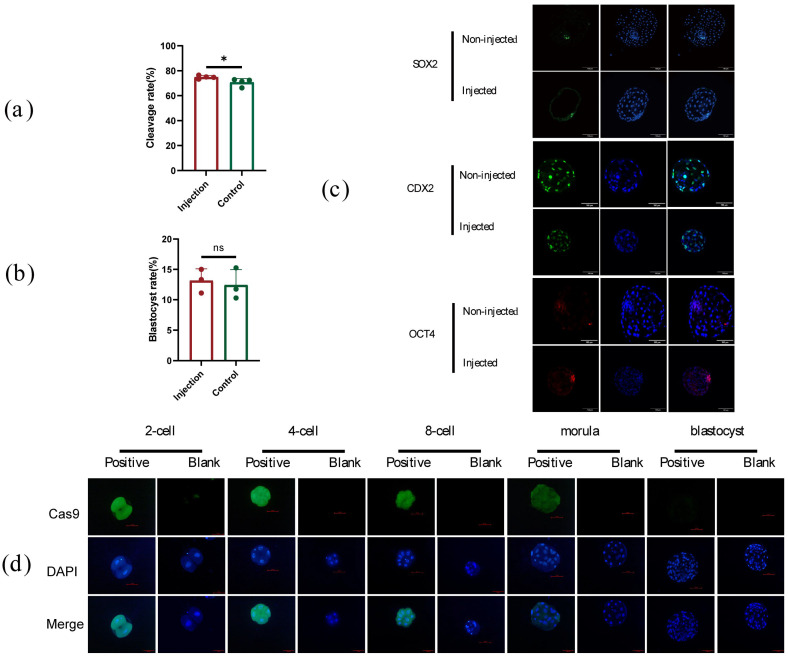
Effect of microinjection on embryo development. (**a**) Forty-eight-hour cleavage rate injected and non-injected embryos; (**b**) Blastocyst rate of injected and non-injected embryos; (**c**) Immunofluorescence of pluripotent gene in injected embryos, the ns means not significant; (**d**) Immunofluorescence of Cas9 protein in injected embryos. All data are presented as the mean ± SD. * *p* < 0.05. Scale bar = 100 µm.

**Figure 7 ijms-26-01065-f007:**
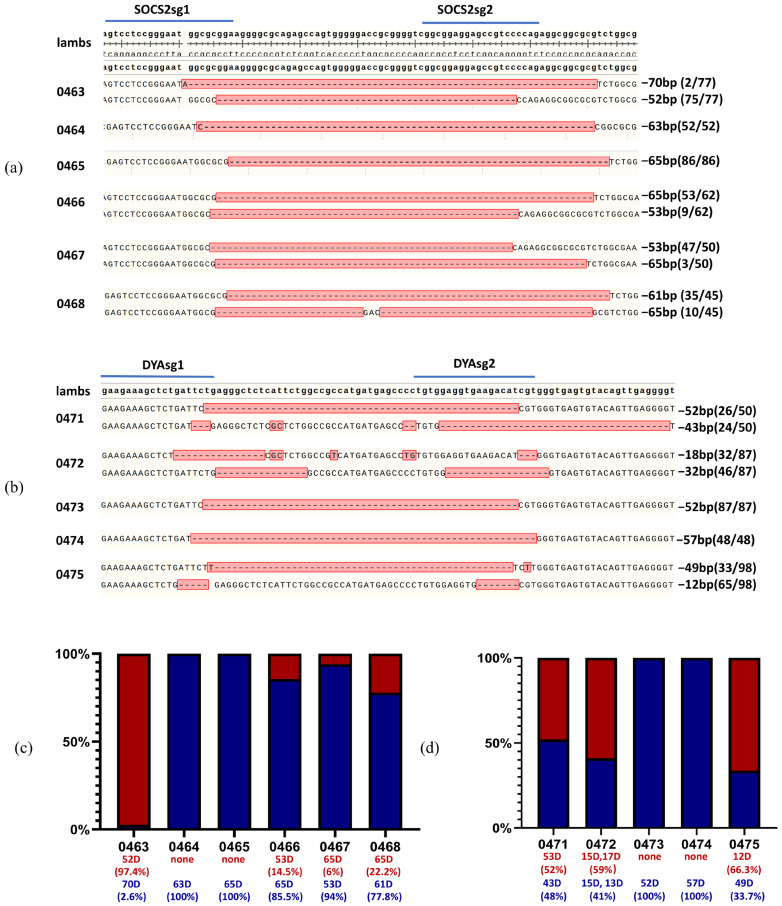
Genotypes of *SOCS2-* and *DYA*-edited sheep. (**a**) Genotypes of *SOCS2* in ear tissues of edited sheep by TA cloning and Sanger sequence, the edited bases are marked in red; (**b**) Genotypes of *DYA*-edited sheep by TA cloning and Sanger sequencing. Numbers of colonies with the genotypes are shown in brackets, the edited bases are marked in red; (**c**) Stacked bar chart of genotypes of gene-modified lambs based on TA cloning results of *SOCS2*-edited lambs; (**d**) Stacked bar chart of genotypes of gene-modified lambs based on TA cloning results of *DYA* edited lambs. Genotypes and ratios were added below the chart with corresponding color. D, deletion.

**Figure 8 ijms-26-01065-f008:**
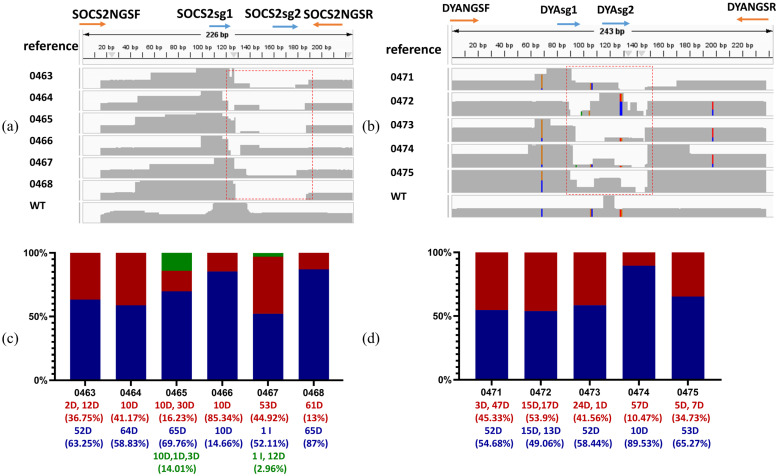
Genotypes of lambs based on NGS sequencing. (**a**) Short reads alignments by BWA-MEM algorithm of *SOCS2*-edited lambs. (**b**) Short reads alignments by BWA-MEM algorithm of *DYA*-edited lambs. Disrupted fragments of target genes were marked as red square. SNPs are marked in colors. (**c**) Stacked bar chart of genotypes based on NGS sequencing results of *SOCS2*-edited lambs. (**d**) Stacked bar chart of genotypes based on NGS sequencing results of *DYA*-edited lambs. Genotypes and ratios were added below the chart with corresponding color. D, deletion; I, Insertion.

**Table 1 ijms-26-01065-t001:** Generation of gene-edited sheep by optimized editing system.

Target Gene	Zygotes	Transplanted Zygotes	Recipient	Pregnant Recipients	Delivered Lambs	Edited Lambs	Ratio of Lambs with Detectable Gene Editing *
*SOCS2*	58	47	7	5	6	6	100%
*DYA*	135	90	15	4	5	5	100%

* The proportion of lambs exhibiting gene editing compared to the total number of lambs.

**Table 2 ijms-26-01065-t002:** Predicted animo acids sequence of target genes in edited sheep.

Target Gene	Lambs	cDNA Sequences	Genotypes	Predicted Animo Acids Sequences	Animo Acids Number
*SOCS2*	WT	---GGGAATGGCGCGGAAGGG------CCAGAGGCGGCGCGTCTGGCG--------CAGGTATAA	/	---GNGAEG---PEAARLA------QV*	198aa
	0463a	---GGGAATGGCGC---------------------CCAGAGGCGGCGCGTCTGGCG------ATATGA	−52 bp	---GNGAQRRRVWR---I*	35aa
	0463b	---GGGAATA-----------------------------------------------------------TCTGG----------ATATGA	−71 bp	---GNIW--------------I*	29aa
	0464	---GGGAATC--------------------------------------------CGGCG--------------------------CAGGTATAA	−64 bp	---GNPA------------------------QV*	177aa
	0465	---GGGAATGGCGCG------------------------------------------------TCTGGC------TGTTAA	−65 bp	---GNGASG--------C*	33aa
	0466a	---GGGAATGGCGCG------------------------------------------------TCTGGC------TGTTAA	−65 bp	---GNGASG--------C*	33aa
	0466b	---GGGAATGGCGC-----------------------CAGA------GACTGTTAA	−53 bp	---GNGAR---------------DC*	37aa
	0467a	---GGGAATGGCGC-----------------------CAGA------GACTGTTAA	−53 bp	---GNGAR---------------DC*	37aa
	0467b	---GGGAATGGCGCG------------------------------------------------TCTGGC------TGTTAA	−65 bp	---GNGASG--------C*	33aa
	0468a	---GGGAATGGCGCG------------------------------------------------TCTGGC------TGTTAA	−65 bp	---GNGASG--------C*	33aa
	0468b	---GGGAATGGCG-----------------GAC---------------------GCGTC------GAAATATGA	−61 bp	---GNGGRV------EI*	32aa
*DYA*	WT	ATGAAGAAAGCTCTGATTCTG------ATCGTGGCGGAC----------------------AGGTGA	/	MKKALIL------IVAD----------R*	288aa
	0471a	ATGAAGAAAGCTCTGATTC----------------------CG----------ACCTGA	−52 bp	MKKALIP----------T*	42aa
	0471b	ATGAAGAAAGCTCTGAT---GAGGGCTCTCGCTCTGGCCGCCATGATGAGCC--TGTG------C---GTTTGA	−43 bp	MKKALMRALALAAMMSLC----V*	40aa
	0472a	ATGAAGAAAGCTCT--------CGC--------TG------CAT---GGC------AGGTGA	−18 bp	MKKALA------------------R*	282aa
	0472b	ATGAAGAAAGCTCTGATTCTG--------------GCCGCCATGATGAGCCCCTGTGG--------------C------GTTTGA	−32 bp	MKKALILAAMMSPCG------V*	37aa
	0473	ATGAAGAAAGCTCTGATTC----------------------CG----------ACCTGA	−52 bp	MKKALIP----------T*	42aa
	0474	ATGAAGAAAGCTCTGAT--------------------------GGCG----------------AGGTGA	−57 bp	MKKALMA----------------R*	269aa
	0475a	ATGAAGAAAGCTCTGATTCTT--------------------TCT------AGGTGA	−49 bp	MKKALILS----------------T*	43aa
	0475b	ATGAAGAAAGCTCTG-----GAGGGCTCTCATTCTGGCCGCCATGATGAGCCCCTGTGGAGGTG-------C------AGGTGA	−12 bp	MKKALEGSHSGRHDEPLWRC----R*	284aa

* Stop codons.

**Table 3 ijms-26-01065-t003:** Efficiency of gene editing by microinjection of sheep zygotes in previous studies.

Target Gene	Transplanted Zygotes	Pregnant Recipients/Recipients	Edited Lambs/Delivered Lambs	Editing Efficiency (%)	Editing Tools
*MSTN* [14]	213	31/55	2/35	5.7	Cas9
*MSTN*, *ASIP* and *BCO2* [15]	578	77/82	35/36	97.2	Cas9
*FGF5* [16]	100	14/53	3/18	16.7	Cas9
*ASIP* [17]	92	6/60	5/6	83.3	Cas9
*BMPR1B* [18]	279	16/39	7/21	33.3	Cas9
*SOCS2* [8]	53	3/8	3/4	75	CBE (Cytosine base editor)
*FecB* [19]	95	6/18	6/8	75	ABE (adenine base editing)
*MSTN* [20]	345	14/58	8/16	50	Cas9
*TBXT* [11]	338	31/216	19/28	67.9	Cas9&ssODN
*FecB* [21]	122	15/45	5/7	71.4	PE (prime editing)
*TBXT* [21]	140		3/8	37.5	
*MSTN* [22]	70	5/13	9/10	90	Cas9
*MSTN/FGF5* [23]	1201	78/236	12/64	18.8	Cas9

## Data Availability

The data presented in this study are available upon request from the corresponding author.

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
