# Peer review of "Optimization of CRISPR/Cas9 Gene Editing System in Sheep (Ovis aries) Oocytes via Microinjection"

_ijms, 2025, doi:10.3390/ijms26031065_

Round 1
Reviewer 1 Report
Comments and Suggestions for Authors
Comments about the manuscript:
“Optimization of CRISPR/Cas9 Gene Editing System in Sheep Oocytes via Microinjection”
The use of the CRISPR/Cas9 system for molecular design selection of livestock, combined with microinjection of zygotes, is not yet optimized. The aim of the proposed work was to evaluate the impact of various factors on the efficiency of Cas9 editing in microinjected mature sheep oocytes, compared to mature oocytes. The results led to the improvement of a system for generating genetically modified sheep.
This complex work of validation of a genetically modified oocyte and zygote production system brings interesting and useful results. It could be published after some improvements of the manuscript
Title: specify the scientific name (genus, species and subspecies) of the animals giving the oocyte.
For the whole text: use italics to write the names of genes. (eg: page 6, line 175).
Page 7, figure 6. Add a scale bar on each image.
Provide an image of the negative control or, more simply, give the result obtained with the negative controls in the legend.
Page 8, line 205. “(…) simpler in execution [13], and results (…): there is no reference between 13 and 26. I think the missing references are those in table 2.
Page 9, table Z. “efficiency of gene editing by microinjection of sheep zygotes in previous studies.”: It would be helpful to give the references in brackets (see note above).
Page 11, line 302. “Cumulus-oocyte complexes (COCs) were separated from purchased ovaries”: Briefly explain the method used. How many COCs were treated for each point?
Page 11, 4.6. Immunofluorescence. Were there any negative controls (e.g. omission of specific antibody)? Explain how these were performed.
I have no particular comments to make on the supplementary file.
Reviewer 2 Report
Comments and Suggestions for Authors
In this MS the authors developed and implemented an optimized microinjection system, which demonstrated significantly enhanced efficiency in sheep gene editing. This system provides a promising approach for generating gene-edited sheep with greatly improved editing success.
The study is well-structured and the results were clearly described in the text.
To the reviewer’s opinion, addressing the following points would enhance clarity and further strengthen the manuscript.
-Line 79 To better understand the improvement, could the authors explain why the maturation rate show 6% increase and the number of suitable oocytes increase by 30%?
-Line 127 Could the authors quantify the GFP expression? high and low is not clear enough. How many injected oocytes are GFP+?
-Line 179 the review suggests to add this comment that could explain tha activity of Cas9 during time: probably the mRNA and the Cas9 protein are left, degraded during time.
-Line 192 the authors should claim at the beginning of the 2.2 section that they are using a couple of gRNA for SOCS 2 and for DYA gene.
-Line 195 : the reviewer suggests to tone down “These results demonstrate that the gene-194 editing system established in this study enables efficient gene editing in sheep” . 90 zygotes treated with Cas system, were transplanted into 15 recipients. only 5 live lambs. I would not claim efficient. See table 2 (SOCS2 ref 19)
-Fig 7 how exactly the authors measured the % of editing? they analysed the live lambs, but which cells they used to extract the genomic DNA?. The authors sequenced few colonies for each lamb (from 8 to 27). They should perform an NGS for each SOCS2 and DYA KO lamb. By using for instance CRISPResso, the authors should evaluate the in frame and out of frame editing and for in frame editing the authors should identified potential stop codon in the coding seq to demonstrate, at least by genomic analysis, that the editing will result in the abrogation of protein expression.
Round 2
Reviewer 2 Report
Comments and Suggestions for Authors
The reviewer thanks the authors for the revision which enhanced clarity and strength of the manuscript. However , I have some more comments, mainly related to the presentation of the NGS data
Major comment: Comment to the response 6: The reviewer thanks for the inclusion of NGS data. However, I have few comments on these data.:
-Please specify in the legend of table 5 the abbreviation I ,D and V ( I guess insertion deletion and variation ? which variation?).
-Table 5 is more interesting than the images in Fig8. a , b which do not show the frequency and the type of indels. I agree to authors which demonstrated the editing in all lambs but the frequency by NGS in table 5 is really variable and does not correlate with Sanger data (see 0463 ,0465, 0468, 0471,0473) reported in fig 7. I strongly suggest the authors to include the frequency and type of NGS data, as WT , deletion insertion , SNP variation, for each lamb in the principle fig 8 as panel c and d using for instance 1 histogram/lamb divided in WT , del, ins ,V and SNP. For those lambs showing different editing by NGS and Sanger , the authors could discuss the requirement of deep sequencing investigation to evaluate editing efficiency and outcome at genomic level , important issues when pursuing a knock out
Minor comment:
-Line 138: the authors could not claim “higher fluorescence intensity had higher gene editing efficiency “ because the difference in fig 3d is ns, thus comparable. Could the authors better explain in the text line 139 what they mean for editing efficiency fig 3d and cleavage efficiency fig 3e?
-Comment to the response 2: since the authors used image J , could they state, in the material and methods, the threshold used to define high and low signals ? this could help to understand the GFP spots in fig S2. Please, define over....... or under.........pixel of whatever, compared to the background.
-Comment to the response 5: instead of “These results showed that all delivered lambs were gene-modified” I would specify “These results showed that all born lambs were gene-modified” simply because the delivery occurred in 15 recipient
Round 3
Reviewer 2 Report
Comments and Suggestions for Authors
The reviewer appreciates the authors' efforts to address concerns.